PERSPECTIVE

# The replication crisis has led to positive structural, procedural, and community changes

Max Korbmacher[1,2,3], Flavio Azevedo[4,35 ✉], Charlotte R. Pennington[5],
Helena Hartmann[6], Madeleine Pownall[7], Kathleen Schmidt[8],
Mahmoud Elsherif[9], Nate Breznau[10], Olly Robertson[11], Tamara Kalandadze[12],
Shijun Yu[9], Bradley J. Baker[13], Aoife O'Mahony[14], Jørgen Ø. -S. Olsnes[15],
John J. Shaw[16], Biljana Gjoneska[17], Yuki Yamada[18], Jan P. Röer[19],
Jennifer Murphy[20], Shilaan Alzahawi[21], Sandra Grinschgl[22],
Catia M. Oliveira[23], Tobias Wingen[24], Siu Kit Yeung[25], Meng Liu[26],
Laura M. König[27], Nihan Albayrak-Aydemir[28,29], Oscar Lecuona[30,31],
Leticia Micheli[32] & Thomas Evans[33,34]

The emergence of large-scale replication projects yielding successful rates substantially lower than expected caused the behavioural, cognitive, and social sciences to experience a so-called 'replication crisis'. In this Perspective, we reframe this 'crisis' through the lens of a credibility revolution, focusing on positive structural, procedural and community-driven changes. Second, we outline a path to expand ongoing advances and improvements. The credibility revolution has been an impetus to several substantive changes which will have a positive, long-term impact on our research environment.

After several notable controversies in 2011[1–3], skepticism regarding claims in psychological science increased and inspired the development of projects examining the replicability and reproducibility of past findings[4]. Replication refers to the process of repeating a study or experiment with the goal of verifying effects or generalising findings across new models

[1] Department of Health and Functioning, Western Norway University of Applied Sciences, Bergen, Norway. [2] NORMENT Centre for Psychosis Research, University of Oslo and Oslo University Hospital, Oslo, Norway. [3] Mohn Medical Imaging and Visualisation Center, Bergen, Norway. [4] Department of Psychology, University of Cambridge, Cambridge, UK. [5] School of Psychology, Aston University, Birmingham, UK. [6] Department of Neurology, University of Essen, Essen, Germany. [7] School of Psychology, University of Leeds, Leeds, UK. [8] Department of Psychology, Ashland University, Ashland, USA. [9] School of Psychology, University of Birmingham, Birmingham, UK. [10] SOCIUM Research Center on Inequality and Social Policy, University of Bremen, Bremen, Germany. [11] Department of Psychiatry, University of Oxford, Oxford, UK. [12] Department of Education, ICT and Learning, Ostfold University College, Halden, Norway. [13] Department of Sport and Recreation Management, Temple University, Philadelphia, USA. [14] School of Psychology, Cardiff University, Cardiff, UK. [15] Kavli Institute for Systems Neuroscience, Norwegian University of Science and Technology, Trondheim, Norway. [16] Division of Psychology, De Montfort University, Leicester, UK. [17] Macedonian Academy of Sciences and Arts, Skopje, North Macedonia. [18] Faculty of Arts and Science, Kyushu University, Fukuoka, Japan. [19] Department of Psychology and Psychotherapy, Witten/Herdecke University, Witten, Germany. [20] Department of Applied Science, Technological University Dublin, Dublin, Ireland. [21] Graduate School of Business, Stanford University, Standford, USA. [22] Institute of Psychology, University of Graz, Graz, Austria. [23] Department of Psychology, University of York, York, UK. [24] Institute of General Practice and Family Medicine, University of Bonn, Bonn, Germany. [25] Department of Psychology, Chinese University of Hong Kong, Hong Kong, China. [26] Faculty of Education, University of Cambridge, Cambridge, UK. [27] Faculty of Life Sciences: Food, Nutrition and Health, University of Bayreuth, Bayreuth, Germany. [28] Open Psychology Research Centre, Open University, Milton Keynes, UK. [29] Department of Psychological and Behavioural Science, London School of Economics and Political Science, London, UK. [30] Department of Psychology, Universidad Rey Juan Carlos, Madrid, Spain. [31] Faculty of Psychology, Universidad Autónoma de Madrid, Madrid, Spain. [32] Institute of Psychology, Leiden University, Leiden, The Netherlands. [33] School of Human Sciences, University of Greenwich, Greenwich, UK. [34] Institute for Lifecourse Development, University of Greenwich, Greenwich, UK. [35] Present address: Department of Social Psychology, University of Groningen, Groningen, The Netherlands. ✉email: flavio.azevedo@rug.nl

or populations, whereas reproducibility refers to assessing the accuracy of the research claims based on the original methods, data, and/or code (see Table 1 for definitions).

In one of the most impactful replication initiative of the last decade, the Open Science Collaboration[5] sampled studies from three prominent journals representing different sub-fields of psychology to estimate the replicability of psychological research. Out of 100 independently performed replications, only 39% were subjectively labelled as successful replications, and on average, the effects were roughly half the original size. Putting these results into a wider context, a minimum replicability rate of 89% should have been expected if all of the original effects were true (and not false positives; ref. [6]). Pooling the Open Science Collaboration[5] replications with 207 other replications from recent years resulted in a higher estimate; 64% of effects successfully replicated with effect sizes being 32% smaller than the original effects[7]. While estimations of replicability may vary, they nevertheless appear to be sub-optimal—an issue that is not exclusive to psychology and found across many other disciplines (e.g., animal behaviour[8–10]; cancer biology[11]; economics[12]), and symptomatic of persistent issues within the research environment[13,14]. The 'replication crisis' has introduced a number of considerable challenges, including compromising the public's trust in science[15] and undermining the role of science and scientists as reliable sources to inform evidence-based policy and practice[16]. At the same time, the crisis has

provided a unique opportunity for scientific development and reform. In this narrative review, we focus on the latter, exploring the replication crisis through the lens of a credibility revolution[17] to provide an overview of recent developments that have led to positive changes in the research landscape (see Fig. 1).

Recent discussions have outlined various reasons why replications fail (see Box 1). To address these replicability concerns, different perspectives have been offered on how to reform and promote improvements to existing research norms in psychological science[18–20]. An academic movement collectively known as open scholarship (incorporating Open Science and Open Research) has driven constructive change by accelerating the uptake of robust research practices while concomitantly championing a more diverse, equitable, inclusive, and accessible psychological science[21,22].

These reforms have been driven by a diverse range of institutional initiatives, grass-roots, bottom-up initiatives, and individuals. The extent of a such impact led Vazire[17] to reframe the replicability crisis as a credibility revolution, acknowledging that the term crisis reflects neither the intense self-examination of research disciplines in the last decade nor the various advances that have been implemented as a result.

Scientific practices are behaviours[23] and can be changed, especially when structures (e.g., funding agencies), environments (e.g., research groups), and peers (e.g., individual researchers)

## Table 1 Terminology adopted from the glossary of Open Science terminology[50].

| Term | Definition |
|---|---|
| Credibility Revolution | The problems and the solutions resulting from a growing distrust in scientific findings, following concerns about the credibility of scientific claims (e.g., low replicability). The term has been proposed as a positive alternative to the term replicability crisis, and includes the many solutions to improve the credibility of research, such as pre-registration, transparency, and replication. |
| Open Science | An umbrella term reflecting the idea that scientific knowledge of all kinds, where appropriate, should be openly accessible, transparent, rigorous, reproducible, replicable, accumulative, and inclusive, all which are considered fundamental features of the scientific endeavour. Open science consists of principles and behaviours that promote transparent, credible, reproducible, and accessible science. Open science has six major aspects: open data, open methodology, open source, open access, open peer review, and open educational resources. |
| Open Scholarship | 'Open scholarship' is often used synonymously with 'open science', but extends to all disciplines, drawing in those which might not traditionally identify as science-based. It reflects the idea that knowledge of all kinds should be openly shared, transparent, rigorous, reproducible, replicable, accumulative, and inclusive (allowing for all knowledge systems). Open scholarship includes all scholarly activities that are not solely limited to research such as teaching and pedagogy. |
| Questionable Research Practices | A range of activities that intentionally or unintentionally distort data in favour of a researcher's own hypotheses—or omissions in reporting such practices—including; selective inclusion of data, hypothesising after the results are known (HARKing), and p-hacking. Popularised by John et al.[202]. |
| Replicability | An umbrella term, used differently across fields, covering concepts of: direct and conceptual replication, computational reproducibility/replicability, generalizability analysis and robustness analyses. Some of the definitions used previously include: a different team arriving at the same results using the original author's artefacts; a study arriving at the same conclusion after collecting new data; as well as studies for which any outcome would be considered diagnostic evidence about a claim from prior research. |
| Replication Crisis | The finding, and related shift in academic culture and thinking, that a large proportion of scientific studies published across disciplines do not replicate (e.g., ref. [5]). This is considered to be due to a lack of quality and integrity of research and publication practices, such as publication bias, questionable research practices and a lack of transparency, leading to an inflated rate of false positive results. Others, starting with Vazire[17], have described this process as a 'Credibility Revolution' towards improving these practices. |
| Reproducibility | A minimum standard on a spectrum of activities ("reproducibility spectrum") for assessing the value or accuracy of scientific claims based on the original methods, data, and code. For instance, where the original researcher's data and computer codes are used to regenerate the results, often referred to as computational reproducibility. Reproducibility does not guarantee the quality, correctness, or validity of the published results. In some fields, this meaning is, instead, associated with the term "replicability" or 'repeatability'. |
| Transparency | Transparency refers to a combination of availability and accountability, or practically, having one's actions open and accessible for external evaluation. Transparency pertains to researchers being honest about theoretical, methodological, and analytical decisions made throughout the research cycle. Transparency can be usefully differentiated into "scientifically relevant transparency" and "socially relevant transparency". While the former has been the focus of early Open Science discourses, the latter is needed to provide scientific information in ways that are relevant to decision makers and members of the public. |

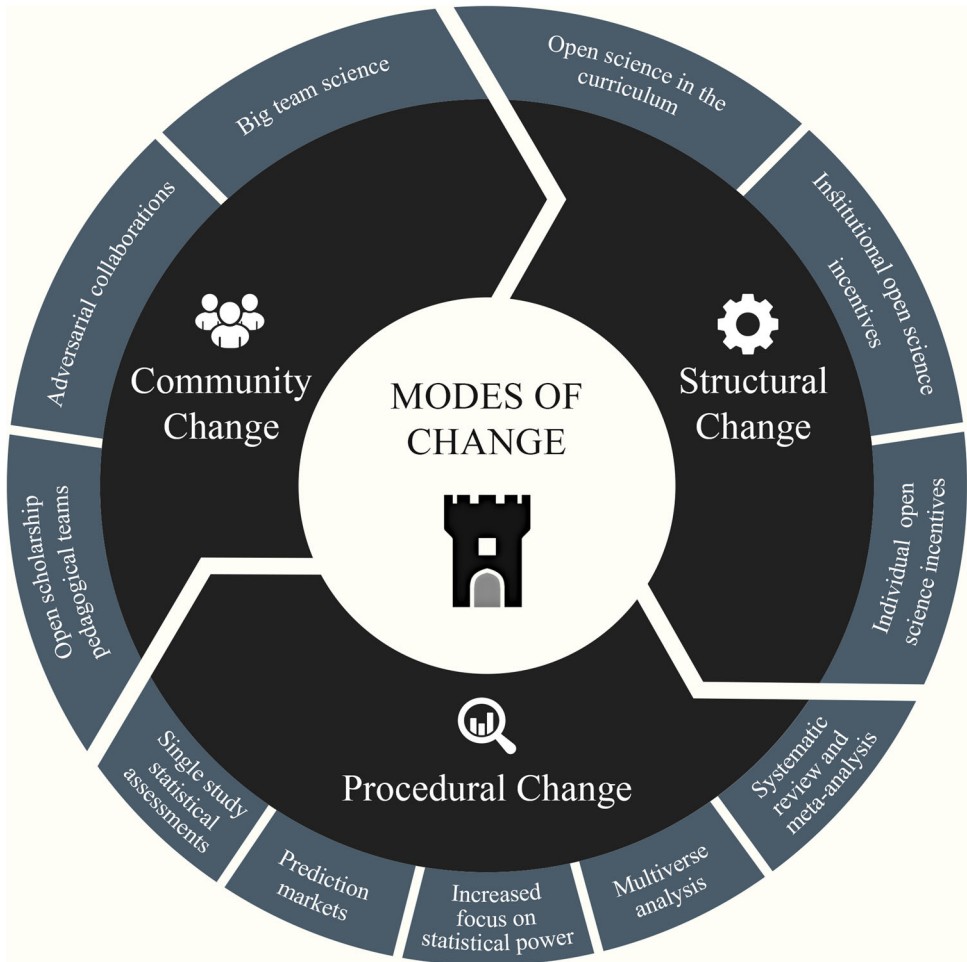

**Fig. 1 Modes of change towards scientific credibility.** This figure presents an overview of the three modes of change proposed in this article: structural change is often evoked at the institutional level and expressed by new norms and rules; procedural change refers to behaviours and sets of commonly used practices in the research process; community change encompasses how work and collaboration within the scientific community evolves.

facilitate and support them. Most attempts to change the behaviours of individual researchers have concentrated on identifying and eliminating problematic practices and improving training in open scholarship[23]. Efforts to change individuals' behaviours have ranged from the creation of grass-roots communities to support individuals to incorporate open scholarship practices into their research and teaching (e.g., ref. [24]) to infrastructural change (e.g., creation of open tools fostering the uptake of improved norms such as the software StatCheck[25] to identify statistical inconsistencies en-masse, providing high-quality and modularized training on the underlying skills needed for transparent and reproducible data preparation and analysis[26] or documenting contributions and author roles transparently[27]).

The replication crisis has highlighted the need for a deeper understanding of the research landscape and culture, and a concerted effort from institutions, funders, and publishers to address the substantive issues. Despite the creation of new open access journals, they still face challenges in gaining acceptance due to the prevailing reputation and prestige-based market. These stakeholders have made significant efforts, but their impact remains isolated and infrequently reckoned. As a result, although there have been positive developments, progress toward a systemic transformation in how science is considered, actioned, and structured is still in its infancy.

In this article, we take the opportunity to reflect upon the scope and extent of positive changes resulting from the credibility revolution. To capture these different levels of change in our complex research landscape, we differentiate between (a) structural, (b) procedural, and (c) community change. Our categorisation is not informed by any given theory, and there are overlaps and similarities across the outlined modes of change. However, this approach allows us to consider change in different domains: (a) embedded norms, (b) behaviours, and (c) interactions, which we believe assists in demonstrating the scope of optimistic changes allowing us to empower and retain change-makers towards further scientific reform.

## Structural change
In the wake of the credibility revolution, structural change is seen as crucial to achieving the goals of open scholarship, with new norms and rules often being developed at the institutional level. In this context, there has been increasing interest in embedding open scholarship practices into the curriculum and incentivizing researchers to adopt improved practices. In the following, we describe and discuss examples of structural change and its impact.

**Embedding replications into the curriculum.** Higher Education instructors and programmes have begun integrating open science practices into the curriculum at different levels. Most notably, some instructors have started including replications as part of basic research training and course curricula[28], and there are freely

**Box 1 | Why research fails to replicate. This box outlines some explanations for the low replicability of research at the individual and structural levels. A more exhaustive overview can be found in ref. [14]**

*Individual level.* Questionable research practices (QRPs) at the individual researcher level may help understand the replicability crisis. Researchers have significant flexibility in processing and analysing data, including the exclusion of outliers and running multiple tests on subsets of the data, leading to false positive results[203,204]. QRPs also involve measurement, such as omitting to report psychometric properties when found to be unsatisfactory, potentially compromising replicability[205,206]), likely with similar negative effects for replicability. Researchers continue to employ a variety of QRPs, which are significant contributors to low replicability[202,204,207-209], and a lack of transparency in reporting can mask and exacerbate these issues[14].
*Structural level.* Characteristics of the academic system also contribute widely to low replicability. For example, misaligned incentives can influence researchers trying to obtain career stability or advancement and encourage the usage of unethical research behaviours[210] such as QRPs[103,211]. More generally, the incentivisation of research is conveyed through common academic aphorisms such as 'publish or perish'[212]. Many of these incentives are driven by research institutions and their governing agencies but also by publishers and funding agencies[213]. For example, the emphasis on arbitrary publication metrics, such as the impact factor[214] or the h-index, create perverse incentives[215] that ultimately reinforce the prioritization of research quantity over quality[85,92,216], with little-to-no requirements for transparency in the publishing process[201]. Such emphasis is particularly evident in publication practices, where novel and hypothesis-supporting research has historically been viewed more positively by editors and reviewers, and thus published more frequently[217]. Many journals prevent or resist publication of null-findings and replications[218] often due to perceived 'lack of contribution', or prioritising novelty over incremental developments[219-221]. This selective publishing creates 'publication bias', a distortion of the literature to over-represent positive findings and under-represent negative ones, giving misleading representations of existing effects[50,78,222]. Negative findings can often end in the 'file-drawer' and are never published[223].
*Challenges with replications.* While replicability is an essential feature of scientific research, its application can vary depending on the research field, with standardization and control of covariates varying for different objects of study[200]. Moreover, replication is not conclusive in ensuring scientific quality due to the heterogeneity of the concept[224]. Differences in replicability can be attributed to several factors such as variations in procedures, measurement characteristics, or evidence strength between original studies and their replications[7,225-227]. Additionally, the characteristics of the effects themselves may vary in size depending on time or context[228]. However, low reliability and systematic errors in original studies[229], missing formalizations of verbalizations, for example of concepts, definitions, and results[174], or the inappropriate use and interpretation of statistical tests[102,103,170], and weak theoretical development underpinning hypothesized effects[230] can lead to mismatches between statistical tests and their interpretations, ultimately influencing replication rates.

available, curated materials covering the entire process of executing replications with students[29] (see also forrt.org/reversals). In one prominent approach, the Collaborative Replications and Education Project[30,31] integrates replications in undergraduate courses as coursework with a twofold goal: educating undergraduates to uphold high research standards whilst simultaneously advancing the field with replications. In this endeavour, the most cited studies from the most cited journals in the last three years serve as the sample from which students select their replication target. Administrative advisors then rate the feasibility of the replication to decide whether to run the study across the consortium of supervisors and students. After study completion, materials and data are submitted and used in meta-analyses, for which students are invited as co-authors.

In another proposed model[32-34], graduate students complete replication projects as part of their dissertations. Early career researchers (ECRs) are invited to prepare the manuscripts for publication[35,36] and, in this way, students' research efforts for their dissertation are utilised to contribute to a more robust body of literature, while being formally acknowledged. An additional benefit is the opportunity for ECRs to further their career by publishing available data. Institutions and departments can also profit from embedding these projects as these not only increase the quality of education on research practices and transferable skills but also boost research outputs[21,37].

If these models are to become commonplace, developing a set of standards regarding authorship is beneficial. In particular, the question of what merits authorship can become an issue when student works are further developed, potentially without further involvement of the student. Such conflicts occur with other models of collaboration (see Community Change, below; ref. [38]) but may be tackled by following standardized authorship templates, such as the Contributor Roles Taxonomy (CRediT), which helps detail each individual's contributions to the work[27,39,40].

**Wider embedding into curricula.** In addition to embedding replications, open scholarship should be taught as a core component of Higher Education. Learning about open scholarship practices has been shown to influence student knowledge, expectations, attitudes, and engagement toward becoming more effective and responsible researchers and consumers of science[41]. It is therefore essential to adequately address open scholarship in the classroom, and promote the creation and maintenance of open educational resources supporting teaching staff[21,29,37]. Gaining an increased scientific literacy early on may have significant long-term benefits for students, including the opportunity to make a rigorous scientific contribution, acquire a critical understanding of the scientific process and the value of replication, and a commitment to research integrity values such as openness and transparency[24,41-43]. Embedding open research practices into education further shapes personal values that are connected to research, which will be crucial in later stages of both academic and non-academic careers[44]. This creates a path towards open scholarship values and practices becoming the norm rather than the exception. It also links directly to existing social movements often embraced among university students to foster greater equity and justice and helps break down status hierarchies and power structures (e.g., decolonisation, diversity, equity, inclusion and accessibility efforts)[45-49].

Various efforts to increase the adoption of open scholarship practices into the curriculum are being undertaken by pedagogical teams with the overarching goal of increasing research rigour and transparency over time. While these changes are structural, they are often driven by single or small groups of individuals, who are usually in the early stages of their careers and receive little recognition for their contributions[21]. An increasing number of grassroot open science organisations contribute to different educational roles and provide resources, guidelines, and community. The breadth of tasks required in the pedagogic reform towards open scholarship is exemplified by the Framework for Open and Reproducible Research Training (FORRT), focusing on reform and meta-scientific research to advance research transparency, reproducibility, rigour, social justice, and ethics[24]. The FORRT community is currently running more than 15 initiatives which include summaries of open scholarship literature, a crowdsourced glossary of open scholarship terms[50], a

literature review of the impact on students of integrating open scholarship into teaching[41], out-of-the-box lesson plans[51], a team working on bridging neurodiversity and open scholarship[46,52], and a living database of replications and reversals[53]. Other examples of organisations providing open scholarship materials are ReproducibiliTea[54], the RIOT Science Club, the Turing Way[55], Open Life Science [56], OpenSciency[57], the Network of Open Science Initiatives[58], Course Syllabi for Open and Reproducible Methods[59], the Carpentries[60], the Embassy of Good Science[61], the Berkeley Initiative for Transparency in Social Sciences[62], the Institute for Replication[63], Reproducibility for Everyone[64], the International Network of Open Science[65], and Reproducibility Networks such as the UKRN[66].

These and other collections of open-source teaching and learning materials (such as podcasts, how-to guides, courses, labs, networks, and databases) can facilitate the integration of open scholarship principles into education and practice. Such initiatives not only raise awareness for open scholarship but also level the playing field for researchers from countries or institutions with fewer resources, such as the Global South and low- and middle-income countries, referring to the regions outside of Western Europe and North America that are primarily politically and culturally marginalised such as regions in Asia, Latin America, and Africa[51,67].

**Incentives**. Scientific practice has been characterized by problematic rewards, such as prioritizing research quantity over quality and emphasizing statistically significant results. To foster a sustained integration of open scholarship practices, it is essential to revise incentive structures. Current efforts have focused on developing incentives that target various actors, including students, academics, faculties, universities, funders, and journals[68–70]. However, as each of these actors has different—and sometimes competing—goals, their motivations to engage in open scholarship practices can vary. In the following, we discuss recently developed incentives that specifically target researchers, academic journals, and funders.

*Targeting researchers.* Traditional incentives for academics to advance in their career are publishing articles, winning grants, and signalling the quality of the published work (e.g., perceived journal prestige)[45,71]. In some journals, researchers are now given direct incentives for the preregistration of study plans and analyses before study execution, and for openly sharing data and materials in the form of (open science) badges, with the aim of signalling study quality[23]. However, the extent to which badges can be used to increase open scholarship behaviours remains unclear; while one study[72] reports increased data sharing rates among articles published in Psychological Science with badges, a recent randomized control trial shows no evidence for badges increasing data sharing[73], suggesting that more effective incentives or complementary workflows are required to motivate researchers to engage in open research practices[74].

Furthermore, there are incentives provided for different open scholarship practices, such as using the Registered Report publishing format[75,76]. Here, authors submit research protocols for peer-review before data collection or analyses (in the case of secondary data). Registered Reports meeting high scientific standards are given provisional acceptance ('in-principle acceptance') before the results are known. Such format shifts the focus from the research outcomes to methodological quality and realigns incentives by providing researchers with the certainty of publication when adhering to the preregistered protocol[75,76]. Empirical evidence has also found that Registered Reports are perceived to be higher in research quality than regular articles, as

well as equivalent in creativity and importance[77], while also allowing to report more negative results[78], which may provide further incentives for researchers to adopt this format.

*Targeting journals and funders.* Incentives are not limited to individual researchers but also to the general research infrastructure. One example of this is academic journals, which are attempting to implement open science standards to remain current and competitive by significantly increasing open publishing options and formulating new guidelines reinforcing and enforcing these changes[79]. For example, the Center for Open Science introduced the Transparency and Openness Promotion (TOP) Guidelines[80] comprising eight modular standards to reflect journals' transparency standards. Namely, citation standards, data transparency, analytic methods transparency, research materials transparency, design and analysis transparency, study pre-registration, analysis plan preregistration, and replication. Building on these guidelines, the TOP factor quantifies the degree to which journals implement these standards, providing researchers with a guide on selecting journals accordingly. Based on TOP and other Open Science best-practices, there are also guides for editors available (e.g., ref. [81]). Similarly, organisations such as NASA[82], UNESCO[83], and the European Commission[84] all came to support open scholarship efforts publicly, and on an international level. There are moreover efforts to open up funding options through the Registered Reports funding schemes[85]. Here, funding allocation and publication review are being combined into a single process, reducing both the burden on reviewers and opportunities for questionable research practices. Finally, large-scale policies are being implemented supporting open scholarship practices, such as Plan-S[86,87], mandating open publishing when the research is funded by public grants. The increase in open access options illustrates how journals are being effectively incentivized to expand their repertoire and normalize open access[88]. At the same time, article processing charges have increased, causing new forms of inequities between the haves and have-nots, and the exclusion of researchers from low-resource universities across the globe[88,89]. As Plan-S shapes the decision space of journals and researchers, it is an incentive with the promise of long-term change.

Several initiatives aim to re-design systems such as peer review and publishing. Community peer reviews (e.g., PeerCommunityIn[90]) is a relatively new system in which experts review and recommend preprints to journals. Future developments in the direction of community peer review might contain an increased usage of overlay journals, meaning that the journals themselves do not manage their own content (including peer review) but rather select and curate content. The peer review procedures can also be changed, as shown by the recent editorial plans in the journal e-Life to abolish accept/reject decisions during peer review[91], and as reflected by a recommendation-based system of the community peer review system.

Evaluation of researchers in academic settings has historically been focused on the quantity of papers they publish in high-impact journals[45], despite criticism of the impact factor metric[71], and their ability to secure grants[85,92]. In response to this narrow evaluation, a growing number of research stakeholders, including universities, have signed declarations such as the San Francisco Declaration of Research Assessment (DORA) or the agreement by the Coalition for Advancing Research Assessment (CORA). This initiative aims to broaden the criteria used for hiring, promotion, and funding decisions by considering all research outputs, including software and data, and considering the qualitative impact of research, such as its policy or practice implications. To promote sustained change towards open scholarship practices, some institutions have also modified the requirements for hiring

committees to consider such practices[69] (see, for example, the Open Hiring Initiative[93]). Such initiatives incentivize researchers to adopt open scholarship principles for career advancement.

### Procedural change
Procedural change refers to behaviours and sets of commonly used practices in the research process. We describe and discuss prediction markets, statistical assessment tools, multiverse analysis, and systematic reviews and meta-analysis as examples of procedural changes.

**Prediction markets of research credibility**. In recent years, researchers have employed prediction markets to assess the credibility of research findings[94–99]. Here, researchers invite experts or non-experts to estimate the replicability of different studies or claims. Large prediction market projects such as the repliCATS project have yielded replicability predictions with high classification accuracy (between 61% and 86%[96,97]). The repliCATS project implemented a structured, iterative evaluation procedure to solicit thousands of replication estimates which are now being used to develop prediction algorithms using machine learning. Though many prediction markets are composed of researchers or students with research training, even lay people seem to perform better than chance in predicting replicability[100]. Replication markets are considered both an alternative and complementary approach to replication since certain conditions may favour one approach over the other. For instance, replication markets may be advantageous in cases where data collection is resource-intensive but less so when study design is especially complex. Therefore, replication markets offer yet another tool for researchers to assess the credibility of existing and hypothetical works. In that sense, it is an ongoing discussion whether low credibility estimates from replication markets can be used to inform decisions on which articles to replicate[101].

**Statistical assessment tools**. Failure to control error rates and design high-power studies can contribute to low replication rates[102,103]. In response, researchers have developed various quantitative methods to assess expected distributions of statistical estimates (i.e., $p$-values), such as p-curving[104], z-curving[105], and others. P-curve assesses publication bias by plotting the distribution of $p$-values across a set of studies, measuring the deviation from an expected uniform distribution of $p$-values considering a true null hypothesis[104]. Like p-curve, the z-curve assesses the distribution of test statistics while considering the power of statistical tests and false discovery rate within a body of literature[105]. Additionally, and perhaps most importantly, such estimations of bias in the literature identify selective reporting trends and help establish a better estimate of whether replication failures may be due to features of the original study or features of the replication study. Advocates of these methods argue for decreasing $\alpha$-levels (i.e., the probability of finding a false positive/committing a type I error) when the likelihood of publication bias is high to allow for increased power and confidence in findings. Other researchers have called for reducing $\alpha$-levels for all tests (e.g., from 0.05 to 0.005[106]), rethinking null hypothesis statistical testing (NHST) and considering exploratory NHST[107], or abandoning NHST altogether[108] (see for an example, ref. [109]). However, these approaches are not panaceas and are unlikely to address all the highlighted concerns[19,110]. Instead, researchers have recommended simply justifying the alpha for tests with regard to the magnitude of acceptable Type I versus Type II (false negative) errors[110]. In this context, equivalence testing[111] or Bayesian analyses[112] have been proposed as suitable approaches to directly assess evidence for the alternative hypothesis against

evidence for the null hypothesis[113]. Graphical user interface (GUI) based statistical software packages, like JASP[114] and Jamovi[115], have played a significant role in making statistical methods such as equivalence tests and Bayesian statistics accessible to a broader audience. The promotion of these methods, including practical walkthroughs and interactive tools like Shiny apps[111,112], has further contributed to their increased adoption.

**Single-study statistical assessments**. A range of useful tools has been developed to pursue open values. For example, the accuracy of reported findings may be assessed by running simple, automated error checks, such as StatCheck[25]. Validation studies[25] reported high sensitivity (larger than 83%), specificity (larger than 96%), and accuracy (larger than 92%) of this tool. Other innovations include the Granularity-Related Inconsistency of Means (GRIM) test[116] aiming to evaluate the consistency of mean values of integer data (e.g., from Likert-type scales), considering sample size and the number of items. Another is the Sample Parameter Reconstruction via Iterative TEchniques (SPRITE), which reconstructs samples and estimates of the item value distributions based on reported descriptive statistics[117]. Adopting these efforts can serve as an initial step in reviewing existing literature, to ensure that findings are not the result of statistical errors or potential falsification. Researchers themselves can implement these tools to check their work and identify potential errors. With greater awareness and use of such tools, we can increase accessibility and enhance our ability to identify unsubstantiated claims.

**Multiverse analysis**. The multitude of researcher degrees of freedom—i.e., decisions researchers can make when using data—have been shown to influence the outcomes of analyses performed on the same data[118,119]. In one investigation, 70 independent research teams analysed the same nine hypotheses with one neuroimaging dataset, and results show data cleaning and statistical inferences varied considerably between teams: no two groups used the same pipeline to pre-process the imaging data, which ultimately influenced both results and inference drawn from the results[118]. Another systematic effort comprising of 161 researchers in 73 teams independently investigated and tested a hypothesis central to an extensive body of scholarship using identical cross-country survey data—that more immigration will reduce public support for government provision of social policies—revealing a hidden universe of uncertainty[120]. The study highlights that the scientific process involves numerous analytical decisions that are often taken for granted as nondeliberate actions following established procedures but whose cumulative effect is far from insignificant[119]. It also illustrates that, even in a research setting with high accuracy motivation and unbiased incentives, reliability between researchers may remain low, regardless of researchers' methodological expertise[119]. The upshot is that idiosyncratic uncertainty may be a fundamental aspect of the scientific process that is not easily attributable to specific researcher characteristics or analytical decisions. However, increasing transparency regarding researcher degrees of freedom is still crucial, and multiverse analyses provide a useful tool to achieve this. By considering a range of feasible and reasonable analyses, researchers can test the same hypothesis across various scenarios and determine the stability of certain effects as they navigate the often large 'garden of forking paths'[121]. Multiverse analyses (and their sibling, sensitivity analyses) can now be performed more often to provide authoritative evidence of an effect on substantive research questions[122–127]. Such an approach helps researchers to determine the robustness of a finding by pooling evidence from a range of appropriate analyses.

**Systematic review and meta-analysis.** Systematic reviews or meta-analyses are used to synthesise findings from several primary studies[128,129], which can reveal nuanced aspects of the research while keeping a bird's-eye perspective, for example, by presenting the range of effect sizes and resulting power estimates. Methods have been developed to assess the extent of publication bias in meta-analyses, and, to an extent, correct for it, using methods such as funnel plot asymmetry tests[130]. However, there are additional challenges influencing the results of meta-analyses and systematic reviews and hence their replicability, such as researcher degrees of freedom in determining inclusion criteria, methodological approaches, and the rigour of the primary studies[131,132]. Thus, researchers have developed best practices for open, and reproducible systematic reviews and meta-analyses such as Preferred Reporting Items for Systematic Reviews and Meta-Analyses (PRISMA)[133], Non-Interventional, Reproducible, and Open (NIRO)[134] systematic review guidelines, the Generalized Systematic Review Registration Form[135], and PROSPERO, a register of systematic review protocols[136]. These guides and resources provide opportunities for more systematic accounts of research[137]. These guidelines often include a risk of bias assessment, where different biases are assessed and reported[138]. Yet, most systematic reviews and meta-analyses do not follow standardized reporting guidelines, even when required by the journal and stated in the article, reducing the reproducibility of primary and pooled effect size estimates[139]. An evaluated trial of enhanced requirements by some journals as part of the submission process[140] did lead to a slowly increasing uptake in such practices[141], with later findings indicating that protocol registrations increased the quality of associated meta-analyses[142]. Optimistically, continuous efforts to increase transparency appear to have already contributed to researchers more consistently reporting eligibility criteria, effect size information, and synthesis techniques[143].

## Community change

Community change encompasses how work and collaboration within the scientific community evolves. We describe two of these recent developments: Big Team Science and adversarial collaborations.

**Big Team Science.** The credibility revolution has undoubtedly driven the formation and development of various large-scale, collaborative communities[144]. Community examples of such approaches include mass replications, which can be integrated into research training[30,31,34,145], and projects conducted by large teams and organisations such as the Many Labs studies[146,147], the Hagen Cumulative Science Project[148], the Psychological Science Accelerator[149], and the Framework for Open and Reproducible Research Training (FORRT)[24].

A promising development to accelerate scientific progress is Big Team Science—i.e., large-scale collaborations of scientists working on a scholarly common goal and pooling resources across labs, institutions, disciplines, cultures, and countries[14,150,151]. Replication studies are often the focus of such collaborations, with many of them sharing their procedural knowledge and scientific insights[152]. This collaborative approach leverages the expertise of a consortium of researchers, increases research efficiency by pooling resources such as time and funding, and allows for richer cross-cultural samples to draw conclusions from[150,153]. Big Team Science emphasizes various practices to improve research quality, including interdisciplinary internal reviews, incorporating multiple perspectives, implementing uniform protocols across participating labs, and recruiting larger and more diverse samples[41,50,149,150,152,154]. The latter also extends to researchers themselves; Big Team Science can increase representation, diversity, and equality and allow researchers to collaborate by either coordinating data collection efforts at their respective institutions or by funding the data collection of researchers who may not have access to funds[155].

Big-Team Science represents a prime opportunity to advance open scholarship goals that have proven to be the most difficult to achieve, including diversity, equity, inclusion, accessibility, and social justice in research. Through a collaborative approach that prioritizes the inclusion of disenfranchised researchers, coalition building, and the redistribution of expertise, training, and resources from the most to the least affluent, Big Team Science can contribute to a more transparent and robust science that is also more inclusive, diverse, accessible, and equitable. This participatory approach creates a better and more just science for everyone[52]. However, it is important to critically examine some of the norms, practices, and culture associated with Big Team Science to identify areas for improvement. Big Team Science projects are often led by researchers from Anglo-Saxon and Global North institutions, while the contributions of researchers from the Global South are oftentimes diluted in the ordering of authors—i.e., authors from Global North tend to occupy positions of prestige such as the first, corresponding, and last author (e.g., refs. [146,147,156–162]) while researchers from Low- and Middle-income countries are compressed in the middle. Moreover, there are also challenges associated with collecting data in low-and-middle-income countries that are often not accounted for, such as limited access to polling infrastructure or technology and the gaping inequities in resources, funding, and educational opportunities. Furthermore, journals and research institutions do not always recognize contributions to Big Team Science projects, which can unequally negatively impact the academic careers of already marginalized researchers. For example, some prominent journals prefer mentioning consortium or group names instead of accommodating complete lists of author names in the byline, with the result that immediate author visibility decreases (see for example, ref. [156]). To promote social justice in Big Team Science practices, it is crucial to set norms that redistribute credit, resources, funds, and expertise rather than preserving the status quo of extractive intellectual labour. These complex issues must be examined carefully—and now—to ensure current customs do not unintendedly sustain colonialist, extractivist, and racist research practices that pervade academia and society at large. Big Team Science stakeholders must see to it that, as a minimum, Big Team Science doesn't perpetuate existing inequalities and power structures.

**Adversarial collaborations.** Scholarly critique typically occurs after research has been completed, for example, during peer-review or in back-and-forth commentaries of published work. With some exceptions[161,163–166], rarely do researchers who support contradictory theoretical frameworks work together to formulate research questions and design studies to test them. 'Adversarial collaborations' of this kind are arguably one of the most important developments in procedures to advance research because they allow for a consensus-based resolution of scientific debates and they facilitate more efficient knowledge production and self-correction by reducing bias[3,167]. An example is the Transparent Psi Project[168] which united teams of researchers both supportive and critical of the idea of extra-sensory perception, allowing for a constructive dialogue and more agreeable consensus in conclusion.

A related practice to adversarial collaborations is that of 'red teams', which can be applied by both larger and smaller teams of researchers playing 'devil's advocate' between one another. Red

teams work together to constructively criticise each other's work or to find errors during (but preferably early in) the entire research process, with the overarching goal of maximising research quality (Lakens, 2020). By avoiding errors "before it is too late", red teams have the potential to save large amounts of resources[150]. However, whether these initiatives contribute to research that is less biased in its central assumptions depends on wherein the "adversarial" nature lies. For example, two researchers may hold the same biased negative views towards one group, but opposing views on the implications of group membership on secondary outcomes. An adversarial collaboration from the same starting point pitching different methodological approaches and hypotheses about factors associated with group membership would still suffer from the same fundamental biases.

### Expanding structural, procedural and community changes
To expand the developments discussed and to address current challenges in the field, we now highlight a selection of areas that can benefit from the previously described structural, procedural, and community changes, namely: (a) generalizability, (b) theory building, and (c) open scholarship for qualitative research, and (d) diversity and inclusion as an area necessary to be considered in the context of open scholarship.

**Generalizability**. In extant work, the generalizability of effects is a serious concern (e.g., refs. [169,170]). Psychological researchers have traditionally focused on individual-level variability and failed to consider variables such as stimuli, tasks, or contexts over which they wish to generalise. While accounting for methodological variation can be partially achieved through statistical estimation (e.g., including random effects of stimuli in models) or acknowledging and discussing study limitations, unmeasured variables, stable contexts, and narrow samples still present substantive challenges to the generalizability of results[170].

Possible solutions may lay in Big-team science and large-scale collaborations. Scientific communities such as the Psychological Science Accelerator (PSA) have aimed to test the generalizability of effects across cultures and beyond the Global North (i.e., the affluent and rich regions of the world, for example, North America, Europe, and Australia[149]). However, Big Team Science projects tend to be conducted voluntarily with very few resources in order to understand the diversity of a specific phenomenon (e.g., ref. [171]). The large samples required to detect small effects may make it difficult for single researchers from specific countries to achieve adequate power for publication. Large global collaborations, such as the PSA, can therefore contribute to avoiding wasted resources by conducting large studies instead of many small-sample studies[149]. At the same time, large collaborations might offer a chance to counteract geographical inequalities in research outputs[172]. However, such projects also tend to recruit only the most accessible (typically student) populations from their countries, thereby potentially perpetuating issues of representation and diversity. Yet, increased efforts of international teams of scientists offer opportunities to provide both increased diversity in the research team and the research samples, potentially increasing generalisability at various stages of the research process.

**Formal theory building**. Researchers have suggested that the replication crisis is, in fact, a "theory crisis"[173]. Low rates of replicability may be explained in part by the lack of formalism and first principles[174]. One example is the improper testing of theory or failures to identify auxiliary theoretical assumptions[103,170]. The verbal formulation of psychological theories and hypotheses cannot always be directly tested with

inferential statistics. Hence, generalisations provided in the literature are not always supported by the used data. Yarkoni[170] has recommended moving away from broad, unspecific claims and theories towards specific quantitative tests that are interpreted with caution and increased weighting of qualitative and descriptive research. Others have suggested formalising theories as computational models and engaging in theory testing rather than null hypothesis significance testing[173]. Indeed, many researchers may not even be at a stage where they are ready or able to test hypotheses[175]. Additional discussion of improving psychological theory and its evaluation is needed to advance the credibility revolution. Hence, a suggested approach to solving methodological problems is to (1) define variables, population parameters, and constants involved in the problem, including model assumptions, to then (2) for-mulate a formal mathematical problem statement. Results are (3) used to interrogate the problem. If the claims are valid, (4a) examples can be used to present practical relevance, while also (4b) presenting possible extensions and limitations. Finally, (5) policy making recommendations can be given. Additional discussion of improving psychological theory and its evaluation is needed to advance the credibility revolution. Such discussions reassessing the application of statistics (in the context of statistical theory) are important steps in improving research quality[174].

**Qualitative research**. Open scholarship research has focused primarily on quantitative data collection and analyses, with substantively less consideration for compatibility with qualitative or mixed methods[51,176–178]. Qualitative research presents methodological, ontological, epistemological, and ethical challenges that need to be considered to increase openness while preserving the integrity of the research process. The uniqueness, context-dependent, and labour-intensive features of qualitative research can create barriers, for example, to preregistration or data sharing[179,180]. Similarly, some of the tools, practices, and concerns of open scholarship are simply not compatible with many qualitative epistemological approaches (e.g., a concern for replicability; ref. [41]). Thus, a one-size-fits-all approach to qualitative or mixed methods data sharing and engagement with other open scholarship tools may not be appropriate for safeguarding the fundamental principles of qualitative research (see review[181]). However, there is a growing body of literature offering descriptions on how to engage in open scholarship practices when executing qualitative studies to move the field forward[6,179,182–184], and protocols are being developed specifically for qualitative research, such as preregistration templates[185], practices increasing transparency[186], and curating[187] and reusing qualitative data[188]. Better representation of the application of open scholarship practices like a buffet, which can be chosen from, depending on the projects and its limitations and opportunities[6,189] is ongoing. Such an approach is reflected in various studies describing the tailored application of open scholarship protocols in qualitative studies[182,184].

It is important to note that qualitative research also has dishes to add to the buffet of open science[178,190]. Qualitative research includes practices that realize forms of transparency that currently lack in quantitative work. One example is the practice of reflexivity, which aims to make transparent the positionality of the researcher(s) and their role in the production and interpretation of the data (see e.g., refs. [191–193]). A different form of transparency is 'member checking'[194], which makes the participants in a study part of the analysis process by asking them to comment on a preliminary report of the analysis. These practices—e.g., member checking, positionality, reflexivity, critical team discussions, external audits and others—would likely

hold potential benefits for strictly quantitative research by promoting transparency and contextualization[178,193].

Overall, validity, transparency, ethics, reflexivity, and collaboration can be fostered by engaging in qualitative open science; open practices which allow others to understand the research process and its knowledge generation are particularly impactful here[179,183]. Irrespective of the methodological and epistemological approach, then, transparency is key to the effective communication and evaluation of results from both quantitative and qualitative studies, and there have been promising developments within qualitative and mixed research towards increasing the uptake of open scholarship practices.

**Diversity and inclusion.** An important point to consider when encouraging change is that the playing field is not equal for all actors, underlining the need for flexibility that takes into account regional differences and marginalised groups as well as differences in resource allocation when implementing open science practices[51]. For example, there are clear differences in the availability of resources by geographic region[195,196] and social groups, by ethnicity[197] or sex and gender[51,198,199]. Resource disparities are also self-sustaining as, for instance, funding increases the chances of conducting research at or beyond the state-of-the-art which in turn increases the chances of obtaining future funding[195]. Choosing (preferably free) open access options, including preprints and post-prints is one step allowing scholars to access resources irrespective of their privileges. An additional possibility is to waive article processing charges for researchers from low, or low-and-middle income countries. Other options are pooled funding applications, re-distributions of resources in international teams of researchers, and international collaborations. Big Team Science is a promising avenue to produce high-quality research while embracing diversity[150]; yet, the predominantly volunteering-based system of such team science might exclude researchers who do not have allocated hours or funding for such team efforts. Hence, beyond these procedural and community changes, structural change aiming to foster diversity and inclusion is essential.

**Outlook: what can we learn in the future?**
Evidenced by the scale of developments discussed, the replication crisis has motivated structural, procedural, and community changes that would have previously been considered idealistic, if not impractical. While developments within the credibility revolution were originally fuelled by failed replications, these in themselves are not the only issue of discussion within the credibility revolution. Furthermore, replication rates alone may not be the best measure of research quality. Instead of focusing purely on replicability, we should strive to maximize transparency, rigour, and quality in all aspects of research[18,200]. To do so, we must observe structural, procedural, and community processes as intertwined drivers of change, and implement actionable changes on all levels. It is crucial that actors in different domains take responsibility for improvements and work together to ensure that high-quality outputs are incentivized and rewarded[201]. If one is fixed without the other (e.g., researchers focus on high-quality outputs [*individual level*] but are incentivised to focus on novelty [*structural level*]), then the problems will prevail, and meaningful reform will fail. In outlining multiple positive changes already implemented and embedded, we hope to provide our scientific community with hope, and a structure, to make further advances in the crises and revolutions to come.

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

## Acknowledgements

We want to thank Ali H. Al-Hoorie for his contribution with reference formatting and commenting on the manuscript, and Sriraj Aiyer, Crystal Steltenpohl, Maarten Derksen, Rinske Vermeij, and Esther Plomp for the comments provided on an earlier version of the manuscript. No funding has been received for this work.

## Author contributions

Author contributions and roles are laid out based on the CRediT (Contributor Roles Taxonomy). Please visit https://www.casrai.org/credit.html for details and definitions of each of the roles listed below[27]. Conceptualization: M.K., F.A. and T.R.E. Methodology: M.K., F.A. and T.R.E. Project administration: M.K. and F.A. Visualisation: M.K. and F.A. Writing—original draft: M.K., F.A., T.R.E. C.R.P. Writing—review and editing: M.K., F.A., H.H., M.M.E., N.B., A.O.M., T.K., J.Ø.S.O., C.R.P., J.J.S., M.P., B.G., Y.Y., J.P.R., J.M., S.A., B.J.B., S.G., C.M.O., T.W., S.K.Y., M.L., L.M.K., N.A.A., O.L., T.R.E., O.R., L.M., K.S. and S.Y.

## Competing interests

This work is as an initiative from The Framework for Open and Reproducible Research Training (FORRT; https://forrt.org). All authors are advocates of open scholarship with different open science organisational affiliations, including FORRT: M.K., F.A., H.H., M.M.E., A.O.M., B.S., T.K., C.R.P., J.J.S., Y.Y., J.P.R., J.M., S.A., B.J.B., S.G., C.M.O., T.W., S.K.Y., M.L., L.M.K., N.A.A., O.L., T.E., O.R., L.M., K.S. and S.Y. ReproducibiliTEA journal clubs: Y.Y., J.S., C.R.P., M.M.E. and O.L. NASA TOPs: F.A. and S.A. Student Initiative for Open Science: M.K. Open Applied Linguistics: M.L. Psychological Science Accelerator: M.K., F.A., T.K., C.R.P., M.P., K.S., M.E., N.B., B.B., A.O.M., B.G., Y.Y., J.R.P., S.A., S.K.Y., M.L., N.A.A., T.E. and M.M.E. ManyMany: B.G. and M.M.E. ManyBabies: M.M.E. The United Kingdom Reproducibility Network: F.A., C.R.P., T.E., O.R., C.R.P. and A.O.M. Reproducible Research Oxford: O.R. The Sports Science Replication Centre: J.M. Norway's Reproducibility Network: T.K. and M.K. Interessengruppe Offene und Reproduzierbare Forschung: H.H. Arbeitsgruppe Open Science der Fachgruppe Gesundheitspsychologie in der Deutschen Gesellschaft für Psychologie: L.M.K. Reproducible Interpretable Open Transparent (RIOT) Science Club: T.K. Center for Open Science: S.A. Open Sciency: F.A. and S.A. Society for the Improvement of Psychological Science: F.A. and S.A. Stanford Center for Open and Reproducible Science: S.A.
