## [Peer Review File · Communications Psychology]

24th Feb 23

Dear Flavio,

Thank you for your patience during the peer-review process. Your manuscript titled "What has been learned from the replication crisis? Towards structural, procedural and community change." has now been seen by 2 reviewers, and I include their comments at the end of this message.

The reviewers are in principle enthusiastic about your work. However, they also mention a number of concerns. Before we can move towards publication of your manuscript in *Communications Psychology*, we ask to to comprehensively address these concerns.

To aid you with that task, I have included a marked-up version of your manuscript. This marked-up version contains pointers relating to the referees requests, as well as editorial feedback, especially with regard to the flow of your argument, the support derived from individual references, the length of the text, and other presentational issues.

In sum, we invite you to revise your manuscript taking into account all reviewer and editor comments.

EDITORIAL POLICIES AND FORMATTING

You will find a complete list of formatting requirements following this link:
<https://www.nature.com/documents/commsj-style-formatting-checklist-review-perspective.pdf>

Please use the checklist to prepare your manuscript for resubmission.

* **TRANSPARENT PEER REVIEW:** *Communications Psychology* uses a transparent peer review system. This means that we publish the editorial decision letters including Reviewers' comments to the authors and the author rebuttal letters online as a supplementary peer review file. We publish these records for all accepted manuscripts. However, on author request, confidential information and data can be removed from the published reviewer reports and rebuttal letters prior to publication. If your manuscript has been previously reviewed at another journal, those Reviewers' comments would not form part of the published peer review file.

If you have any questions about any of our policies or formatting, please don't hesitate to contact me.

Please use the following link to submit your revised manuscript and a point-by-point response to the referees' comments (which should be in a separate document to any cover letter):

[link redacted]

We hope to receive your revised paper within 12 weeks; please let us know if you aren't able to submit it within this time so that we can discuss how best to proceed. If we don't hear from you, and the revision process takes significantly longer, we may close your file.

Please do not hesitate to contact me if you have any questions or would like to discuss these revisions further. We look forward to seeing the revised manuscript and thank you for the opportunity to review your work.

Best wishes
Marike

Marike Schiffer, PhD
Chief Editor
Communications Psychology

REVIEWERS' EXPERTISE:

Reviewer #1: Open Science, Reproducibility

Reviewer #2: Open Science, Reproducibility

REVIEWERS' COMMENTS:

Reviewer #1 (Remarks to the Author):

The paper offers a thorough review of important facets of the Open Scholarship movement in the empirical social sciences. As such, it summarizes relevant changes towards increased credibility and robustness of research. Further, the authors structure these developments, suggesting the differentiation between structural, procedural and community-driven changes. Capitalizing on this differentiation, they propose next steps for solidifying the advance of Open Scholarship.

The proposal of these three modes of change, as well as the discussion based on them, is an important contribution to current debates in the social sciences. Such a meta-perspective on the discourse is a helpful tool for structuring ideas and developing further plans for action. I believe this framework has the potential to influence how researchers think about the developments and future avenues of the Open Scholarship movement.

Although the review of the development of the Open Scholarship movement as well as the reframing as a credibility revolution is certainly an impressively well-put summary, I believe the strength of the paper lies in the proposal of a framework of three modes of change towards increased robustness and credibility of research, and the discussions following from it. I would suggest focusing the manuscript less on the review of the Open Scholarship

movement, and giving more credit to other summary contributions in the published body of literature that already contain descriptions of this process. This would give the authors more space for sharpening the focus of the commentary on developing the three modes framework. While I find the distinction of the three modes of change intuitively plausible, a more developed argument for why these categories were chosen, how they relate to and are distinct from each other, as well as how they were synthesized from the developments of the Open Scholarship movement should be included in a revised version of the paper.

Such a careful layout of the basic structure of the arguments would further strengthen the conclusions already drawn, and perhaps even open the gateways for further ideas stemming from the framework. Considering additional steps that could be taken to bring Open Scholarship to the heart of research practice and culture could include rethinking peer review, publishing and employment structures, and go far beyond. A revision of the paper could outline additional areas where developments could be predicted – or at least discussed.

In addition to these conceptual comments, I have three remarks regarding specifics in the manuscript:

I thought the collection of Grassroots initiatives in Table 2 was a helpful resource. However, it would be great to learn more about how the initiatives displayed therein were selected. Although it is surely unreasonable to expect an exhaustive list of all initiatives providing educational materials about Open Scholarship, I immediately thought about the materials collected by the Network of Open Science Initiatives (NOSI, <https://osf.io/tbkzh/>) or the collection of Course Syllabi for Open and Reproducible Methods (<https://osf.io/vkhbt/>). I imagine other people involved in the Open movement would also intuitively look for a mention of initiatives they are familiar with. Providing a rationale for the in- and exclusions from the list might help readers understand better what the aim of including the list in the paper is.

I feel the manuscript is a bit heavy on citations of prior work by the authors themselves, which is completely understandable (given the incentive systems). Nevertheless, to substantiate the claims made, it would be great if the manuscript referenced more examples of work from other sources.

Finally, I found the section on Open Scholarship in qualitative research surprising, because it seems to miss out on referring to the budding work that is indeed being done in this area to develop Open research practices and promote their use, while taking into account the specific requirements and challenges of such work.

For transparency, I'd like to sign my review: I'm Rima-Maria Rahal.

Reviewer #2 (Remarks to the Author):

Korbamcher et al. conduct a narrative review on "What is learned from the replication crisis?". In their manuscript they analyze scientific opportunities emerging from the replication "crisis" on a structural, procedural and community level.

The manuscript is well written and structured. The reviewed literature is extensive but conveys a largely optimistic view of the credibility revolution. In parts, a more nuanced view could benefit the manuscript. I outline some more specific points below that could be addressed

1. This is a narrative review or perspective. This needs to be mentioned in the title or at least in the abstract.

2. l. 191 "Open Scholarship is predominantly driven by grass-roots bottom up initiatives"

There are many institutional infrastructures that are by no means grass roots (repositories like Zenodo, OSF, EOSC) or teaching programs (FOSTER) or Institutions (e.g. Open Science Offices at many Universities). I recommend to tone this down.

3. l 214 I cannot see how this citation is adding to the point you are making here.

4. What is the motivation to look at the three levels structures, procedures, communities?

How are these defined? How are they different from each other. This needs to be explained.

5. Incentives (in structural change)

You describe positive effects of batches but fail to mention that a randomized control trial has yielded no benefit here. <https://doi.org/10.1098/rsos.191818>

Perhaps it is worthwhile to mention CoARA here as well <https://coara.eu/>

I find that this whole section (incentives) is not connected well to the credibility revolution. Why do we need incentives? How could they look like? What different incentives exist and how should they change did they change through the revolution? As it is I find this part interesting but too shallow to convince the reader that the credibility revolution changes anything here.

6. Nudging: There is no concrete study or trial that is actually testing this. It does not add much to understand the structural changes in the credibility revolution. I suggest writing more in depth on the incentives and omit this section.

7. Statistical power analyses

l.339 I am not sure whether p-curves estimate the power of a statistical test. Rather would say p curves measure the deviation from an expected uniform distribution of p-values | $H_0=TRUE$.

l. 343 Sotola 2022: I do not share your enthusiasm for this preprint and think the claims in the preprint are overblown and z curves are not suited to actually measure what the author promises.

l.355-357 How have they become more accessible? Understanding Bayesian statistics has not become more or less complicated than it was before. Equivalence testing often needs (depending on the bounds) larger samples and is part of clinical studies for ages. It is not like Lakens discovered it (what he also does not claim). You could mention JASP as a tool for accessibility.

8. Qualitative research

Why do you not mention mixed methods here? I feel that this part could use a broader perspective that goes beyond experimental psychology.

There is a template for preregistering qualitative research:

<https://doi.org/10.1080/08989621.2019.1580147>

I sign my reviews

Ulf Toelch BIH QUEST Center

REVIEWERS' COMMENTS:

Reviewer #1 (Remarks to the Author):

The paper offers a thorough review of important facets of the Open Scholarship movement in the empirical social sciences. As such, it summarizes relevant changes towards increased credibility and robustness of research. Further, the authors structure these developments, suggesting the differentiation between structural, procedural and community-driven changes. Capitalizing on this differentiation, they propose next steps for solidifying the advance of Open Scholarship.

The proposal of these three modes of change, as well as the discussion based on them, is an important contribution to current debates in the social sciences. Such a meta-perspective on the discourse is a helpful tool for structuring ideas and developing further plans for action. I believe this framework has the potential to influence how researchers think about the developments and future avenues of the Open Scholarship movement.

Response: Thank you for your positive comments; we are glad that you agree that our article makes an important contribution to current debates in open scholarship in the social sciences. We hope that this article will spark progressive discussions both in research, and the educational landscape.

Although the review of the development of the Open Scholarship movement as well as the reframing as a credibility revolution is certainly an impressively well-put summary, I believe the strength of the paper lies in the proposal of a framework of three modes of change towards increased robustness and credibility of research, and the discussions following from it. I would suggest focusing the manuscript less on the review of the Open Scholarship movement, and giving more credit to other summary contributions in the published body of literature that already contain descriptions of this process. This would give the authors more space for sharpening the focus of the commentary on developing the three modes framework. While I find the distinction of the three modes of change intuitively plausible, a more developed argument for why these categories were chosen, how they relate to and are distinct from each other, as well as how they were synthesized from the developments of the Open Scholarship movement should be included in a revised version of the paper.

Response: We have shortened some areas of the manuscript to refocus our work as suggested, whilst also including an 'info box' for readers who may be less familiar or new to replications, which contains important overviews of the literature. We now provide definitions of the modes of change in the beginning of each section (and in the legend of Figure 1) and include a substantive overview in the background section to outline the rationale of these three modes and their distinctions. We also provide additional references throughout to signpost the seminal work in the area.

Such a careful layout of the basic structure of the arguments would further strengthen the conclusions already drawn, and perhaps even open the gateways for further ideas stemming from the framework. Considering additional steps that could be taken to bring Open Scholarship to the heart of research practice and culture could include rethinking peer review, publishing and employment structures, and go far beyond. A revision of the paper could outline additional areas where developments could be predicted – or at least discussed.

Response: We have now expanded the manuscript in several places to include the discussion around peer review, publishing and employment structures. For example, the following is now included in the "Structural Changes" section under "Embedding replications into the curriculum":

“If these models are to become commonplace, developing a set of standards regarding authorship is beneficial. In particular, the question of what merits authorship can become an issue when student works are further developed, potentially without further involvement of the student. Such conflicts occur with other models of collaboration (see Community Change, below; [46]) but may be tackled by following standardized authorship templates, such as the Contributor Roles Taxonomy (CRediT), which helps detail each individual’s contributions to the work [28, 47, 48].” P.6

The Discussion section now provides further outlooks:

“Several efforts aim to re-design systems such as peer review and publishing. Community peer reviews (e.g., [85]) is a relatively new system in which experts review and recommend preprints to journals. Future developments in the direction of community peer review might contain an increased usage of overlay journals, meaning that the journals themselves do not manage their own content (including peer review) but rather select and curate content. The peer review procedures can also be changed, as shown by the recent editorial decision in E-life to abolish accept/reject decisions during peer review [86], and as reflected by a recommendation-based system of the community peer review system.” P.9

In addition to these conceptual comments, I have three remarks regarding specifics in the manuscript:

I thought the collection of Grassroots initiatives in Table 2 was a helpful resource. However, it would be great to learn more about how the initiatives displayed therein were selected. Although it is surely unreasonable to expect an exhaustive list of all initiatives providing educational materials about Open Scholarship, I immediately thought about the materials collected by the Network of Open Science Initiatives (NOSI, <https://osf.io/tbkzh/>) or the collection of Course Syllabi for Open and Reproducible Methods (<https://osf.io/vkhbt/>). I imagine other people involved in the Open movement would also intuitively look for a mention of initiatives they are familiar with. Providing a rationale for the in- and exclusions from the list might help readers understand better what the aim of including the list in the paper is.

Response: Thank you for this helpful comment. We have now removed Table 2 with the resources as they represent only a small fraction of the available resources and are therefore not a fair representation. Furthermore, a static table would quickly become outdated. Instead, we have taken your suggestions above and have added the recommended links which give an overview of important materials in-text.

I feel the manuscript is a bit heavy on citations of prior work by the authors themselves, which is completely understandable (given the incentive systems). Nevertheless, to substantiate the claims made, it would be great if the manuscript referenced more examples of work from other sources.

Response: We now ensure that all claims are substantiated by relevant scholarship, including additional references, where applicable. At the same time, a lot of the work being done in this space is through FORRT, an initiative made up of several hundred members. Naturally, some of the citations adding to this growing body of work will be from the members of FORRT. However, we have reviewed these again to ensure they are the most appropriate citation to provide.

Finally, I found the section on Open Scholarship in qualitative research surprising, because it seems to miss out on referring to the budding work that is indeed being done in this area to develop Open research practices and promote their use, while taking into account the specific requirements and challenges of such work.

Response: We have now revised this section to outline the growing body of literature on open scholarship for qualitative research. Please find the re-worked section on p. 15-16.

For transparency, I'd like to sign my review: I'm Rima-Maria Rahal.

Response: Thank you for your positive and constructive review which has allowed us to improve our manuscript substantially.

Reviewer #2 (Remarks to the Author):

Korbamcher et al. conduct a narrative review on "What is learned from the replication crisis?". In their manuscript they analyze scientific opportunities emerging from the replication "crisis" on a structural, procedural and community level.

The manuscript is well written and structured. The reviewed literature is extensive but conveys a largely optimistic view of the credibility revolution. In parts, a more nuanced view could benefit the manuscript. I outline some more specific points below that could be addressed

Response: Thank you for your positive comments on our manuscript. We have now revised our manuscript to provide a more nuanced view, in parts, and outline our responses to your additional comments below.

1. This is a narrative review or perspective. This needs to be mentioned in the title or at least in the abstract.

Response: In line with the Editor's comments and guidelines of the journal, the title has now been changed to: "The replication crisis has led to positive structural, procedural, and community changes". In line with your comment, we have amended the Abstract to state that this is a narrative review, as follows:

"The emergence of large-scale replication projects yielding successful rates substantially lower than expected caused the behavioural, cognitive, and social sciences to experience a so-called 'replication crisis'. In this narrative review, we reframe this 'crisis' through the lens of a credibility revolution, focusing on positive structural, procedural and community-driven changes. Second, we outline a path to expand ongoing advances and improvements. The credibility revolution has been an impetus to several substantive changes which will have a positive, long-term impact on our research environment." P.2

2. 1. 191 "Open Scholarship is predominantly driven by grass-roots bottom up initiatives" There are many institutional infrastructures that are by no means grass roots (repositories like Zenodo, OSF, EOSC) or teaching programs (FOSTER) or Institutions (e.g. Open Science Offices at many Universities). I recommend to tone this down.

Response: Thank you for noting this; we have since removed the mentioned sentence, but amended other parts of the manuscript to reflect this feedback, to provide a more nuanced view as you suggest above. One example sentence reads now:

"These reforms have been driven by a diverse range of institutional initiatives, grass-roots, bottom-up initiatives, and individuals." P.5

3. 1 214 I cannot see how this citation is adding to the point you are making here.

Response: The section (including the reference) has now been removed based on the Editor's suggestion.

4. What is the motivation to look at the three levels structures, procedures, communities? How are these defined? How are they different from each other. This needs to be explained.

Response: This is an excellent comment. The rationale for using structural, procedural and community changes as a framework and some explanation is now provided in the end of the “Background” section:

“Scientific practices are behaviours [24] and can be changed, especially when structures (e.g., funding agencies), environments (e.g., research groups), and peers (e.g., individual researchers) facilitate and support it. Most attempts to change the behaviours of individual researchers have concentrated on identifying and eliminating problematic practices and improving training in open scholarship [24]. Efforts to change individuals’ behaviours have ranged from the creation of grass-roots communities to support individuals to incorporate open scholarship practices into their research and teaching (e.g., [25]) to infrastructural change (e.g., creation of open tools fostering the uptake of improved norms such as the software StatCheck [26] to identify statistical inconsistencies en-masse, providing high-quality and modularized training on the underlying skills needed for transparent and reproducible data preparation and analysis [27] or documenting contributions and author roles transparently [28]). The replication crisis has highlighted the need for a deeper understanding of the research landscape and culture, and a concerted effort from institutions, funders, and publishers to address the substantive issues. Despite the creation of new open journals, they still face challenges in gaining acceptance due to the prevailing reputation and prestige-based market. These stakeholders have made significant efforts, but their impact remains isolated and infrequently reckoned. As a result, although there have been positive developments, there has been little progress toward a systemic transformation in how science is considered, actioned, and structured. In this article, we take the opportunity to reflect upon the scope and extent of positive changes resulting from the credibility revolution. To capture these different levels of change in our complex research landscape, we differentiate between a) structural, b) procedural, and c) community change. Our categorisation is not informed by any given theory, and there are overlaps and similarities across the outlined modes of change. However, this approach allows us to consider change in different domains: a) embedded norms, b) behaviours, and c) interactions, which we believe assists in demonstrating the scope of optimistic changes allowing us to empower and retain change-makers towards further scientific reform.” P.5

Furthermore, in the beginning of each of the sections on structural, procedural and community changes we now define these terms:

“In the wake of the credibility revolution, structural change is seen as crucial to achieving the goals of Open Scholarship, with new norms and rules often being developed at the institutional level. In this context, there has been increasing interest in embedding open scholarship practices into the curriculum and incentivizing researchers to adopt improved practices. In the following, we describe and discuss examples of structural change and its impact.” P.6

“Procedural change refers to single behaviours and sets of commonly used practices in the research process. We describe and discuss prediction markets, statistical assessment tools, multiverse analysis, and systematic reviews and meta-analysis as examples of procedural changes.” P.9

“Community change encompasses how work and collaboration within the scientific community transforms. We describe two of these developments in the following: big team science and adversarial collaborations.” P.12

5. Incentives (in structural change)

You describe positive effects of batches but fail to mention that a randomized control trial has yielded no benefit here. <https://doi.org/10.1098/rsos.191818>

Perhaps it is worthwhile to mention CoARA here as well <https://coara.eu/>

I find that this whole section (incentives) is not connected well to the credibility revolution. Why do we need incentives? How could they look like? What different incentives exist and how should they change did they change through the revolution? As it is I find this part interesting but too shallow to convince the reader that the credibility revolution changes anything here.

Response: We have added the randomized controlled trial on badges to the discussion in the “Incentives” section:

“However, the extent to which badges can be used to increase open scholarship behaviours remains unclear; while one study [73] reports increased data sharing rates among articles published in Psychological Science with badges, a recent randomized control trial shows no evidence for badges increasing data sharing [74], suggesting that more effective incentives or complementary workflows are required to motivate researchers to engage in open research practices [75].” (P.8)

We added CoARA to the references at suitable place in the text:

“Current efforts have focused on developing incentives that target various actors, including students, academics, faculties, universities, funders, and journals [69–71].”

Finally, we also re-structured and expanded the section addressing why incentives are useful, what they can look like, and how they developed through the credibility revolution distinguishing between journals and researchers (P.8-9).

6. Nudging: There is no concrete study or trial that is actually testing this. It does not add much to understand the structural changes in the credibility revolution. I suggest writing more in depth on the incentives and omit this section.

Response: We have implemented several changes in the previous nudging section and merged it with the “Incentives” section (which has also been restructured). We have also removed all statements about nudging. Please see the restructured section on P.8-9.

7. Statistical power analyses

1.339 I am not sure whether p-curves estimate the power of a statistical test.

Rather would say p curves measure the deviation from an expected uniform distribution of p-values | $H_0 = \text{TRUE}$.

Response: We agree with this point and have revised the text accordingly:

“*P*-curve assesses publication bias by plotting the distribution of *p*-values across a set of studies, measuring the deviation from an expected uniform distribution of *p*-values considering a true null-hypothesis [98].” P.10

1. 343 Sotola 2022: I do not share your enthusiasm for this preprint and think the claims in the preprint are overblown and z curves are not suited to actually measure what the author promises.

Response: We have removed the preprint reference and revised the sentence. We now refer to the original z-curve article:

“Like *p*-curve, the *z*-curve assesses the distribution of test statistics while considering the power of statistical tests and false discovery rate within a body of literature [99].” P.10

1.355-357 How have they become more accessible? Understanding Bayesian statistics has not become more or less complicated than it was before. Equivalence testing often needs (depending on the bounds) larger samples and is part of clinical studies for ages. It is not like Lakens discovered it (what he also does not claim). You could mention JASP as a tool for accessibility.

Response: In accordance with the Reviewer's comments, we specify the following in the text:

“In this context, equivalence testing [105] or Bayesian analyses [106] have been proposed as suitable approaches to directly assess evidence for the alternative hypothesis against evidence for the null hypothesis [107]. Graphical user interface (GUI) based statistical software packages, like JASP [108] and Jamovi [109], have played a significant role in making statistical methods such as equivalence tests and Bayesian statistics accessible to a broader audience of scientists. The promotion of these methods, including practical walkthroughs and interactive tools like Shiny apps [105, 106], has further contributed to their increased adoption.” P.10-11

8. Qualitative research

Why do you not mention mixed methods here? I feel that this part could use a broader perspective that goes beyond experimental psychology.

There is a template for preregistering qualitative research:

<https://doi.org/10.1080/08989621.2019.1580147>

Response: We now include this template and have re-worked this section to broaden its scope and include mixed methods:

“Thus, a one-size-fits-all approach to qualitative or mixed methods data sharing and engagement with other open scholarship tools may not be appropriate for safeguarding the fundamental principles of qualitative research (see review [168]). However, there is a growing body of literature offering descriptions on how to engage in open scholarship practices when executing qualitative studies to move the field forward [7, 167, 169–171], and protocols are being developed specifically for qualitative research, such as preregistration templates [172]. Better representation of the application of open scholarship practices like a buffet, which can be chosen from, depending on the projects and its limitations and opportunities [7, 173] is ongoing. Such an approach is reflected in various studies describing the tailored application of open scholarship protocols in qualitative studies [169, 171]. Overall, validity, transparency, ethics, reflexivity, and collaboration can be fostered by engaging in qualitative open science; open practices which allow others to understand the research process and its knowledge generation are particularly impactful here [167, 170]. Irrespective of the methodological and epistemological approach, then, transparency is key to the effective communication and evaluation of results from both quantitative and qualitative studies, and there have been promising developments within qualitative and mixed research towards increasing the uptake of open scholarship practices.” P.16

26th Apr 23

Dear Flavio

The revision of your Perspective titled "The replication crisis has led to positive structural, procedural, and community changes" has now been editorially evaluated. Given the comprehensive changes to the work in response to our referees' feedback, I am delighted to say that we are happy, in principle, to publish it in *Communications Psychology* under a Creative Commons 'CC BY' open access license without charge.

We ask you to revise the work one final time. If the revised paper is in *Communications Psychology* format, in accessible style and of appropriate length, we shall accept it for publication immediately. I have attached an edited version of your manuscript, and a checklist to prepare the final revisions. I ask you to attend to each comment in detail.

EDITORIAL REQUESTS:

* Please review the changes in the attached copy of your manuscript, which has been edited for style, and address the comments and queries I have added. If using Word, please use the 'track changes' feature to make the process of accepting your manuscript more efficient.

* Please check whether your manuscript contains third-party images, such as figures from the literature, stock photos, clip art or commercial satellite and map data. If any of the display items in your manuscript (figures, tables, boxes or movies) include images that are the same as, or are adaptations of, previously published images, please fill in the [Third Party Rights Table](http://www.nature.com/licenceforms/sn1/thirdpartyrights-table.doc), and return to us when you submit your revised manuscript. This information will enable us to obtain the necessary rights to re-use such material. If we are unable to obtain the necessary rights to use or adapt any of the material that you wish to use, we will contact you to discuss alternative options.

* *Communications Psychology* uses a transparent peer review system. On author request, confidential information and data can be removed from the published reviewer reports and rebuttal letters prior to publication. If you are concerned about the release of confidential data, please let us know specifically what information you would like to have removed. Please note that we cannot incorporate redactions for any other reasons.

* If you have not done so already, please alert me to any related manuscripts from your group that are under consideration or in press at other journals, or are being written up for submission to other journals (see www.nature.com/authors/editorial_policies/duplicate.html for details).

FORMATTING GUIDELINES:

You will find a complete list of formatting requirements following this link:

<https://www.nature.com/documents/commsj-style-formatting-checklist-review-perspective.pdf>

Please use the checklist to prepare your manuscript for final submission. In the following, I also highlight some issues of particular importance.

** Preface (instead of Abstract)

The paper's preface (up to 100 words; without references) should serve both as a general introduction to the topic, and as a brief, non-technical summary of your main points and their implications. It should start by outlining the background to your article (why the topic is important) and the main question you have addressed, before going on to describe your key points, main conclusions and their general implications. Because we hope that researchers across all fields of psychology will be interested in your work, the preface should be as accessible as possible, explaining essential but specialised terms concisely. We suggest you show your preface to colleagues in other fields to uncover any problematic concepts.

** Length

The ideal length for Review Articles and Perspectives in Communications Psychology is 6,000 words.

** Main text

Please provide three or four section headings in the main text. These should relate to the content of the article rather than being generic. Headings should be no longer than 60 characters (including spaces) and should not use punctuation. Please do not use more than two levels of headings.

** Figures

Please remove all figures from the main text and upload them individually, one figure per file. To ensure the swift processing of your paper please provide the highest quality, vector format, versions of your images (.ai, .eps, .psd) where available. Text and labelling should be in a separate layer to enable editing during the production process. If vector files are not available then please supply the figures in whichever format they were compiled in and not saved as flat .jpeg or .TIFF files. If your artwork contains any photographic images, please ensure these are at least 300 dpi.

* Figures should be simple and informative — multi-part figures are best avoided. Boxes should occupy no more than half a page in the PDF (less than 500 words) and may include a figure.

* References

References appear as superscript Arabic numerals, in order of mention. The reference list mentions references in the numerical order in which they are mentioned in the main text. If a reference is cited more than once, the same number is used throughout the text and the reference receives a single entry in the reference list.

We ask that you select the most significant 5–10% of references in your list for highlighting, and add a single sentence in bold after each of these references to describe the main result and its significance.

Only papers that have been published or accepted by a named publication should be in the reference list (preprints and citations of datasets are also permitted). Unpublished/Submitted research should not be included in the reference list; it should only be mentioned briefly and parenthetically in the main text. Note that no major arguments should rely on unpublished research.

Published conference abstracts and URLs for web sites should be cited parenthetically in the text,

not in the reference list.

Footnotes are not used.

* Competing interests

Please include a "Competing interests" statement after the References. Note that we ask authors to declare both financial and non-financial competing interests. For more details, see <https://www.nature.com/authors/policies/competing.html>. If you have no financial or non-financial competing interests, please state so: "The authors declare no competing interests."

SUBMISSION INFORMATION:

* If you wish, you may also submit a visually arresting image, together with a concise legend, for consideration as a 'Hero Image' on our homepage. The file should be 1400x400 pixels and should be uploaded as 'Related Manuscript File'. In addition to our home page, we may also use this image (with credit) in other journal-specific promotional material.

* Your paper will be accompanied by a two-sentence editor's summary, of between 250-300 characters, when it is published on our homepage. Could you please approve the draft summary below or provide us with a suitably edited version.

Korbmacher and colleagues from the FORRT project discuss how the last decade has seen as a credibility revolution for psychological science, benefitting from structural, procedural and community-driven changes.

In order to accept your paper, we require the following:

* The checklist describing your response to our editorial requests.

* The final version of your text as a Word or TeX/LaTeX file, with any tables prepared using the Table menu in Word or the table environment in TeX/LaTeX and using the 'track changes' feature in Word.

* Production-quality versions of all figures, supplied as separate files. Photographic images should be 300 dpi in RGB format (.jpg, TIFF or native Photoshop format) and any labels/scale bars included in a separate layer from the image. Line art, graphs and schemes should be vector format (.ai, .eps, .pdf); Adobe Illustrator files are preferred and will minimize production time. Any chemical structures or schemes contained within figures should additionally be supplied as separate Chemdraw (.cdx) files.

At acceptance, the corresponding author will be required to complete an Open Access Licence to Publish on behalf of all authors, declare that all required third party permissions have been obtained.

Please note that your paper cannot be sent for typesetting to our production team until we have received this information; **therefore, please ensure that you have this ready when submitting the final version of your manuscript.**

ORCID

Communications Psychology is committed to improving transparency in authorship. As part of our efforts in this direction, we are now requesting that all authors identified as 'corresponding author' create and link their Open Researcher and Contributor Identifier (ORCID) with their account on the Manuscript Tracking System (MTS) prior to acceptance. ORCID helps the scientific community achieve unambiguous attribution of all scholarly contributions. For more information please visit <http://www.springernature.com/orcid>

For all corresponding authors listed on the manuscript, please follow the instructions in the link below to link your ORCID to your account on our MTS before submitting the final version of the manuscript. If you do not yet have an ORCID you will be able to create one in minutes.

IMPORTANT: All authors identified as 'corresponding author' on the manuscript must follow these instructions. Non-corresponding authors do not have to link their ORCIDs but are encouraged to do so. Please note that it will not be possible to add/modify ORCIDs at proof. Thus, if they wish to have their ORCID added to the paper they must also follow the above procedure prior to acceptance.

To support ORCID's aims, we only allow a single ORCID identifier to be attached to one account. If you have any issues attaching an ORCID identifier to your MTS account, please contact the [Platform Support Helpdesk](http://platformsupport.nature.com/).

[link redacted]

We hope to hear from you within two weeks; please let us know if the process may take longer.

Best wishes

Marike

Marike Schiffer, PhD

Chief Editor

Communications Psychology